# Omega-3 Polyunsaturated Fatty Acids—Vascular and Cardiac Effects on the Cellular and Molecular Level (Narrative Review)

**DOI:** 10.3390/ijms23042104

**Published:** 2022-02-14

**Authors:** Ines Drenjančević, Jan Pitha

**Affiliations:** 1Institute and Department of Physiology and Immunology, Faculty of Medicine Osijek, University Josip Juraj Strossmayer, Osijek J. Huttlera 4, HR-31000 Osijek, Croatia; ines.drenjancevic@mefos.hr; 2Scientific Centre of Excellence for Personalized Health Care, University Josip Juraj Strossmayer Osijek, Trg Sv. Trojstva 3, HR-31000 Osijek, Croatia; 3Laboratory for Atherosclerosis Research, Center for Experimental Research, Department of Cardiology, Institute for Clinical and Experimental Medicine, 140 21 Prague, Czech Republic

**Keywords:** omega-3 polyunsaturated fatty acids, cardiovascular disease, eicosapentaenoic acid, docosahexaenoic acid

## Abstract

In the prevention and treatment of cardiovascular disease, in addition to the already proven effective treatment of dyslipidemia, hypertension and diabetes mellitus, omega-3 polyunsaturated fatty acids (n-3 PUFAs) are considered as substances with additive effects on cardiovascular health. N-3 PUFAs combine their indirect effects on metabolic, inflammatory and thrombogenic parameters with direct effects on the cellular level. Eicosapentaenoic acid (EPA) seems to be more efficient than docosahexaenoic acid (DHA) in the favorable mitigation of atherothrombosis due to its specific molecular properties. The inferred mechanism is a more favorable effect on the cell membrane. In addition, the anti-fibrotic effects of n-3 PUFA were described, with potential impacts on heart failure with a preserved ejection fraction. Furthermore, n-3 PUFA can modify ion channels, with a favorable impact on arrhythmias. However, despite recent evidence in the prevention of cardiovascular disease by a relatively high dose of icosapent ethyl (EPA derivative), there is still a paucity of data describing the exact mechanisms of n-3 PUFAs, including the role of their particular metabolites. The purpose of this review is to discuss the effects of n-3 PUFAs at several levels of the cardiovascular system, including controversies.

## 1. Introduction

Manageable vascular and cardiac disorders include mainly atherosclerosis, arteriosclerosis and their fatal complications such as ischemic heart disease, strokes, peripheral artery disease and heart failure. Correction of established cardiovascular risk factors (smoking, dyslipidemias, hypertension, diabetes mellitus) has great potential to be successful in the prevention of these disorders, but many persons and patients still remain at high risk for clinical events. In addition to the already established and in-clinical trials that have been effective in the treatment of dyslipidemia, hypertension and diabetes mellitus, as well as in addition to antithrombotic therapy, omega-3 polyunsaturated fatty acids (n-3 PUFAs) are amongst the most frequently discussed nutritional substances. According to human and experimental studies, they could have robust potential to favorably modify vascular and cardiac pathological processes indirectly through their favorable effects on metabolic parameters but also directly through their effects at the cellular and sub-cellular level [1,2]. In addition, the effect of n-3 PUFAs could also modify important accompanying psychological consequences in, or even causes of, cardiovascular disease, including depressive syndromes in secondary (strokes) [3] and primary prevention, especially in perimenopausal women [4].

Omega-3 fatty acids, also called omega-3 oils, ω-3 fatty acids or n-3 fatty acids, are polyunsaturated fatty acids characterized by the presence of a double bond, three atoms away from the terminal methyl group in their chemical structure. They are widely distributed in nature, being important constituents of lipid metabolism. Three types of omega-3 fatty acids involved in human physiological processes are α-linolenic acid (ALA, C18:3 ω3, found in plant oils), and eicosapentaenoic acid (EPA, C20:5 ω3) and docosahexaenoic acid (DHA, C22:6 ω3), both commonly found in marine fish oils. The main biochemical characteristics of n-3 PUFAs and closely related compounds are shown in Figure 1. Humans are not able to synthesize essential omega-3 fatty acid ALA and can only obtain it through diet. ALA can be used, to a limited extent, when available, to form EPA and DHA by creating additional double bonds along its carbon chain and extending it. ALA (18 carbons and 3 double bonds) is used to make EPA (20 carbons and 5 double bonds), which is then used to make DHA (22 carbons and 6 double bonds). Nevertheless, the ability to make the longer-chain omega-3 fatty acids from ALA is negligible and may be further impaired by aging. In addition, when exposed to air, unsaturated fatty acids could be vulnerable to oxidation, but n-3 PUFAs are used as nutraceuticals in various oxidation-resistant compounds [5]. Regarding the cardiovascular effects of n-3 PUFAs, one extensive meta-analysis [6] and one systematic review [7] concluded that the intake of fish products, including EPA and DHA, has negligible effects on cardiovascular health with the potential exception of ALA, which could moderately reduce the risk of clinical events of cardiovascular disease of atherosclerotic origin [2]. In contrast to these conclusions, recent REDUCE-IT (Reduction of Cardiovascular Events with Icosapent Ethyl–Intervention Trial) and EVAPORATE (Effect of Vascepa on Improving Coronary Atherosclerosis in People With High Triglycerides Taking Statin Therapy) trials [8,9] using EPA-derivative icosapent ethyl at the dose of 4 g/day, described a significant decrease in cardiovascular events and reduction of plaques in coronary arteries. The potential beneficial effects of treatment with n-3 PUFAs were further supported by the open-label, not by placebo controlled Japanese study JELIS (Japan EPA Lipid Intervention Study) published in 2007, which observed a reduction of clinical events in cardiovascular disease of atherosclerotic origin with EPA in hypercholesterolemic patients, both in primary and secondary prevention cohorts [10]. Nevertheless, the ASCEND (A Study of Cardiovascular Events in Diabetes) [11] and STRENGTH (A Long-Term Outcomes Study to Assess Statin Residual Risk Reduction with Epanova in High Cardiovascular Risk Patients with Hypertriglyceridemia) studies [12,13] failed to show any effect of treatment with n-3 PUFAs. Furthermore, another recent study did not show any reduction in clinical events in elderly patients with recent myocardial infarction treated with 1.8 g of n-3 PUFAs daily for 2 years [14]. The relative representation of EPA and DHA and dosage of EPA and its derivatives are hypothesized to be a potential explanation for these differences [15,16]. However, even in the successful REDUCE-IT trial, a significant increase in atrial fibrillation and a non-significant increase in non-serious bleeding episodes were observed, although a low number of such side effects was recorded. In addition, according to a recent meta-analysis, supplementation by n-3 PUFAs could increase the risk of atrial fibrillation [17]. Another concern in successful REDUCE-IT and EVAPORATE trials was the use of a comparator, mineral oil, in the control group with potential proatherogenic potential through elevation of plasma LDL cholesterol, while in other trials with negative results, comparators with a neutral effect on LDL cholesterol were used. In addition, a prospective observational study based on sophisticated calculations in the general population in Copenhagen found that the elevation of LDL cholesterol seen in a control group in the REDUCE-IT trial could increase the risk for cardiovascular events [18]. It should be admitted, however, that as a part of the EVAPORATE study, mineral oil recipients were compared to those receiving cellulose-based placebos, and no difference in plaque progression between both groups was observed [19]; in addition, the extent of LDL cholesterol elevation was regarded as not strongly interfering with results of the REDUCE-IT study.

In summary, trials with n-3 PUFAs differ in composition and dosage, characteristics of participants, duration of the study and, finally, presence and the type of placebo/comparator (Table 1). Despite encouraging making clinical data available, at least for EPA derivatives at high doses, some doubts are still pending [20,21,22]. A better understanding of the biological mechanisms of n-3 PUFAs is, therefore, of importance, including in the large field of immune responses to n-3 PUFAs [23]. Moreover, new approaches to the investigation, including measuring blood levels of n-3 PUFAs with a dedicated n-3 PUFAs index, as suggested by health care professional societies and recent significant scientific data, are to be implemented [24]. In addition, the direct effect of n-3 PUFAs on myocardial tissue could also be of critical importance. The main potential cardiovascular effects of n-3 PUFAs are shown in Figure 2. In this review, we will focus on the biological effects of n-3 PUFAs on multiple levels of cardiovascular systems, including lipoproteins, inflammatory and prothrombogenic factors with an impact on endothelial dysfunction, development and destabilization of atherosclerotic plaques and on clinical manifestations of atherothrombosis.

## 2. The Effects of n-3 PUFAs on Lipid Metabolism

In human studies, the main measurable effect of n-3 PUFA was a reduction of plasma triglycerides. Elevated plasma triglycerides, especially their postprandial concentrations, are considered to be additional risk factors for atherosclerosis [29,30]. The critical point of the n-3 PUFAs effect was the modification of the availability of free/non-esterified fatty acids in the hepatocyte and subsequent change (mainly decrease) in the production of triglycerides. In particular, the increased consumption of n-3 PUFAs reduces the size of very-low-density lipoproteins—precursors of atherogenic LDL particles, at least in some ethnic groups [31]. The postulated mechanism is the increased solubility of VLDL caused by the increased content of phospholipids with n-3 PUFAs on the particle surface detected in women [32], which accelerates lipolysis with an increase of LDL cholesterol concentration [33]. Therefore, on the one hand, the higher intake of n-3 PUFAs is positively associated with total and LDL cholesterol and apolipoprotein B (apoB) concentration in large LDL particles; on the other hand, the proportion of small LDL particles decreases [32]. As small-size-dense LDL particles are more atherogenic, the final effect of n-3 PUFAs intake is, therefore, expected to be atheroprotective. In addition, n-3 PUFAs reduce VLDL production by inhibiting hepatic fatty acid synthesis by decreasing the concentration of free fatty acids in the circulation and through enhancing lipoprotein lipase activity in postprandial status [34]. The positive effect of the increased intake of n-3 PUFAs is associated not only with plasma concentration lipoprotein particles but also with lipoprotein structure; the latter might be more important than previously appreciated.

Both DHA and EPA lower triglyceride concentrations. With plasma concentrations of total cholesterol largely unchanged by EPA and DHA, DHA alone increased LDL cholesterol concentration more than EPA, but as mentioned, it also favorably increases LDL particle size. The effects of EPA and DHA on the inflammatory markers associated with lipoprotein metabolism and glycemic control are inconclusive, with both probably lowering oxidative stress [35]. In successful REDUCE-IT and EVAPORATE trials [8,9], icosapent ethyl treatment significantly and favorably affected non-HDL, LDL- and HDL-cholesterol, apoB and C-reactive protein measured by a highly sensitive method (hsCRP). Interestingly, the favorable clinical effects of active treatment exceeded those expected to result from reducing plasma triglycerides and/or other biomarkers. In addition, a more robust reduction of apoB was observed in successful REDUCE-IT and EVAPORATE trials than in the negative STRENGTH study. However, in the latter study, corn oil, considered to have a neutral impact regarding LDL cholesterol concentration, was used as a comparator. When metabolic effects of icosapent ethyl were compared with mineral oil, the latter increased apoB, LDL cholesterol and hsCRP levels; this effect was not observed with the corn oil in the STRENGTH and OMEMI trials [12,14]. In addition, as already mentioned in a recent observational study from Denmark, the potential adverse effects of changes of LDL cholesterol in the control group exposed to mineral oil were supported [18]. The biological background of the clinically more beneficial effect of EPA, compared to DHA, also comes from experimental studies indicating a more potent antioxidant effect of EPA, not only in various apoB-containing lipoprotein particles (LDL, VLDL and small dense LDL) but also in the cell membranes, irrespectively of the plasma levels of these factors [36,37,38,39], see below. In summary, through the effects of free fatty acids and triglycerides, n-3 PUFAs could favorably attenuate the atherogenic effects of apoB carrying atherogenic lipoproteins [40,41]. Regarding changes of biologically active circulating lipids/lipoproteins, no substantial differences between EPA and DHA were described.

## 3. The Effect of n-3 PUFAs on Inflammatory and Prothrombogenic Processes

Beyond the favorable effects on plasma lipids, n-3 PUFAs are also hypothesized to have a strong impact on the inflammatory and prothrombogenic processes. Several human and experimental studies focused on these factors. One of the most intensively studied areas is the inflammatory status of adipose tissue. Among the most valuable scientific approaches are those presenting a combination of in vivo and in vitro studies. In a study focused on adipose and muscle inflammatory markers in human tissues and evaluating the effect of consumption of fish oils, the effects of n-3 PUFAs on adipocytes and macrophages were additionally analyzed in vitro [42]. In this study, insulin-resistant, nondiabetic subjects were treated with n-3 PUFAs (4 g/day) or placebo for 12 weeks. This treatment reduced plasma macrophage chemoattractant protein 1 (MCP-1) but did not change levels of other cytokines, including adiponectin, interleukin 6 (IL-6), interleukin 10 (IL-10), interleukin 12 (IL-12), tumor necrosis factor-α (TNF-α), resistin, plasminogen activator inhibitor-1 (PAI-1) or leptin. Subjects treated with n-3 PUFAs demonstrated a decrease in the number of adipose tissue macrophages, a decrease in MCP-1 and an increase in the density of capillaries. Moreover, subjects with the most numerous macrophages in adipose tissue demonstrated the greatest response to the treatment. This effect was not observed in skeletal muscles. At the same time, the content of n-3 PUFAs increased after treatment in adipose and muscle tissues; however, no change in insulin sensitivity or adiponectin was detected. Furthermore, in an in vitro study, n-3 PUFAs decreased MCP-1 expression in M1-polarized macrophages (which expressed high levels of MCP-1 inferred to be pro-inflammatory and proatherogenic) but did not affect TNF-α. N-3 PUFAs also attenuated the upregulation of MCP-1 in adipocytes cultured with macrophages. Furthermore, n-3 PUFAs reduced the content of macrophages in adipose tissue, increased capillary density, reduced MCP-1 expression in macrophages of insulin-resistant individuals and also favorably reduced the activity of macrophages and adipocytes in vitro. At the same time, no measurable effect on insulin sensitivity was found. However, in adipose tissue analyzed pre- and posttreatment, several adipose tissue cytokines and chemokines associated with obesity, including IL-6, MCP-1, C–C motif chemokine ligand 2 (CCL2), interleukin 7R and CX3CL1, were markedly less expressed. In a study of obese individuals paralleled by in vitro study of primary human adipocytes (from biopsy specimens) and human THP-1 monocyte-derived macrophages, the effect of the treatment for 8 weeks with a 4 g mixture of EPA and DHA controlled by placebo was analyzed. Both EPA and DHA reduced inflammasome gene expression in adipose tissue and in human adipocytes and macrophages. Moreover, reduced expression of adipose inflammatory genes, including inflammasome-associated IL-18 and IL-1β and circulating IL-18, were detected. In summary, in this study, n-3 PUFA reduced the NLRP3 inflammasome in human adipose tissue by downregulating gene expression in adipocytes and monocytes/macrophages; therefore, they reduced the obesity-related inflammatory status [43]. Another piece of evidence from human studies comes from a randomized double-blinded trial [44], focused on markers of vascular inflammation in 40 healthy volunteers treated by EPA (4 g/day) and using TNF-stimulated human umbilical vein endothelial cells. The main result was significantly reduced gene expression of pro-inflammatory markers including CCL2 by 25%. In addition, significant inverse strong correlations were observed between EPA levels and vascular expression of vascular cell adhesion molecules 1 (VCAM1) (r = −0.56, *p* = 0.001) and CCL2 (r = −0.56, *p* = 0.001) independently of circulating lipids and other inflammatory markers, which were originally low. There was also a trend towards reductions in both VCAM1 and CCL2 with standard fish oil containing predominantly EPA, which increased EPA levels to a similar extent as the sole EPA treatment. However, negative results regarding effects of n-3 PUFAs were presented in an experimental part of this study comparing EPA, DHA (both 600 mg/kg/day), olive oil and no treatment in experimental models of C57Bl/6J mice and ApoE knockout (ApoE−/−) mice fed an atherogenic diet. N-3 PUFAs did not improve plaque burden, morphologic vessel characteristics or lipid burden. A human controlled-randomized clinical trial in young, healthy individuals revealed markers of leukocyte activation (such as CD11a/LFA-1) and antioxidative defense to be related to an increased intake of n-3 PUFAs [45]. Furthermore, n-3 PUFA consumption induced a significant shift towards anti-inflammatory prostanoids and increased levels of pro-resolving oxylipins and increased TGFβ-1 and reduced interleukin 6 secretions by peripheral Th17 cells. Taken together, consumption of n-3 PUFA-enriched-diet favorably altered immune response in regards to inflammation, resolving conditions by affecting lipid mediators and cytokine secretion by T lymphocytes. Another positive effect of n-3 PUFA consumption in humans on the markers of inflammation and oxidative stress was seen in a placebo-controlled, double-blind, randomized, interventional study in young, healthy individuals. In this study, in addition to favorable effects of plasma lipids, significant differences in the markers of leukocyte activation (such as CD11a/LFA-1) and antioxidative defense, following increased intake of n-3 PUFAs in low/intermediate doses (1 g/day n-3 PUFAs) in the form of enriched hen eggs was observed [46]. In this study, endothelial function was also improved, as discussed later. Consumption of n-3 PUFAs significantly improved endothelium-dependent microvascular vasodilation compared to the control group of young, healthy individuals. Furthermore, the plasma concentration of pro-inflammatory cytokine interferon-γ was significantly decreased, and anti-inflammatory interleukin-10 was significantly increased after n-3 PUFA consumption, suggesting favorable anti-inflammatory properties of n-3 PUFAs, which may contribute to enhanced microvascular blood flow [47,48]. In another human study focused on cellular aspects, the effects of EPA, DHA and a mixture of EPA/DHA on the expression of inflammatory genes in THP-1 human macrophages stimulated by lipopolysaccharides were analyzed. Cells were incubated in the presence of n-3 PUFAs and lipopolysaccharides, and messenger ribonucleic acid (mRNA) levels, inflammatory genes and cytokine levels were measured. The mixture of EPA/DHA exerted a better inhibitory effect than either DHA or EPA alone, DHA being more potent than EPA. As for the gene expression, EPA had a greater impact during post-incubation (after lipopolysaccharide treatment), while DHA and EPA/DHA were more potent during co-incubation (after treatment with n-3 PUFAs and lipopolysaccharide). Cytokine concentrations decreased more markedly in the co-incubation condition in stimulated macrophages. The expression of genes involved in inflammation was modified by the dose (10, 50 and 75 μM concentrations), the type of n-3 PUFAs, and the time of incubation [49].

Experimental studies allow for more focus on the pathophysiological aspects of the potential anti-inflammatory effects of PUFAs. Such evidence comes from an experimental study focused on the anti-inflammatory but also on the anti-apoptotic effects of n-3 PUFAs during methotrexate-induced intestinal damage in cell cultures and a rat model. In this study, n-3 PUFAs inhibited NF-κB/COX-2-induced production of pro-inflammatory cytokines and inhibited cell apoptosis, mainly by extrinsic pathways in rats with methotrexate-induced intestinal damage [50]. Strong favorable anti-fibrotic and anti-inflammatory effects of n-3 PUFAs, including modification of transforming growth factor β-1 (TGFβ-1) signaling, PPARα/γ, GPR120, the NLRP3 inflammasome and NF-κB signaling, focused on the effects on the cardiovascular system have been previously reviewed [51]. In contrast to the results from human studies, male mice C57Bl/6J on a high-fat diet with 5% corn oil, a high-fed diet with 40% of the corn oil substituted for purified EPA and DHA triglycerides showed reduced lipid accumulation and levels of the pro-inflammatory fatty acid arachidonic acid in both white and brown adipose tissues, compared with a high-fed diet of corn oil. The transcriptomic analysis showed changes in β-oxidation pathways, supporting the decreased lipid accumulation in the high-fat diet with EPA and DHA. Therefore, data suggest that EPA and DHA supplementation of a high-fat diet may be anti-inflammatory, as well as reduce lipid accumulation in adipose tissues [52].

Closely associated with inflammatory vascular status is the development of relatively benign stages of atherosclerosis to its malignant form—atherothrombosis. The effect of n-3 PUFAs and their metabolites on various components of the hemostatic system, in particular on blood platelets and endothelium, is also of interest. Unfortunately, the results of clinical studies are often contradictory regarding the antithrombotic properties of n-3 PUFAs, and experimental models are not reliable for studying advanced stages of atherosclerosis in humans, which develops over decades. Nevertheless, despite controversies regarding the antithrombotic efficacy of n-3 PUFAs, their use in this indication is still worth investigating. One area of very recent interest is in COVID-19 thrombotic complications. In this respect, a double-blind, randomized clinical trial investigated the effect of n-3 PUFAs (one capsule of 1000 mg omega-3 daily containing 400 mg EPAs and 200 mg DHAs for 14 days) in 128 critically ill patients infected with COVID-19. The intervention group had a significantly higher 1-month survival rate, and supplementation by n-3 PUFAs significantly improved respiratory and renal function [53]. Nevertheless, looking at factors of coagulation, despite the lymphocyte count moderately increasing in the group intervened by n-3 PUFAs compared to the control group, there were no significant between-group differences in the levels of partial thromboplastin time, hematocrit, neutrophil, monocyte, hemoglobin or platelet count. The prothrombotic status could be, in this case, indirectly prevented by kidney function improvement, as indicated by the authors. Therefore, the timing of management by n-3 PUFAs with regard to the actual clinical status of the given individual could also be of importance. However, recently available data did not show the protective effects of icosapent ethyl in COVID-19 [54,55,56].

In general, the potential benefit of shifting up the EPA:arachidonic acid ratio has long been recognized in the prevention of platelet aggregation [57]. The underlying mechanism could be that while cyclooxygenase in circulating platelets converts arachidonic acid to the pro-aggregatory thromboxane-A2, at the same time, it also converts EPA (if present) to anti-aggregatory thromboxane-A3. A similar process was seen in the vessel wall when arachidonic acid is converted to the pro-aggregatory prostacyclin PGI2 while EPA is converted to the anti-aggregatory prostacyclin PGI3. The potential interplay between inflammatory and thrombogenic factors was studied by Dona et al. using resolvin E1 (RvE1)—an EPA-derived lipid mediator generated during the resolution of inflammation and in human vasculature via leukocyte-endothelial cell interactions [58]. RvE1 (in the 10- to 100-nM range for 30 min in human whole blood) rapidly regulated leukocyte expression of adhesion molecules. RvE1 also stimulated L-selectin leakage from monocytes and leukocytes while reducing CD18 expression in both neutrophils and monocytes. On the other hand, RvE1 did not stimulate reactive oxygen species by either neutrophils or monocytes in the whole blood and did not directly stimulate cytokine/chemokine production in heparinized blood. Moreover, in the same study, RvE1 reduced leukocyte rolling by approximately 40% in venules of mice (established by intravital microscopy). In human platelet-rich plasma, RvE1 blocked selectively ADP-stimulated and thromboxane receptor agonist U46619-stimulated platelet aggregation in a concentration-dependent manner. However, RvE1 did not block collagen-stimulated aggregation. These results indicated RvE1 to be a potent modulator of leukocytes and platelet responses. Results of this study also demonstrated novel agonist-specific antiplatelet actions of RvE1, which may explain beneficial actions of EPA through this factor. This study also highlighted the potential interplay between inflammatory and thrombotic processes represented by leucocytes and platelets, respectively, and the potentially favorable effect of n-3 PUFAs on both processes.

Despite assumptions of the pure antithrombotic properties of n-3 PUFAs, reliable evidence on a molecular/cellular level for this effect is still absent and, as recently reviewed by Stupin et al. [59], the impact of dietary n-3 PUFAs on hemorheology and coagulability, as well as microvascular function, is still an under-investigated field. Nevertheless, a huge population study, ARIC (the Atherosclerosis Risk in Communities), focused on this phenomenon, and n-3 PUFA intake (increased fish intake of 1 serving per day) was negatively associated with concentrations of fibrinogen, factor VIII and von Willebrand factor (in black and white individuals) and positively associated with protein C (in white individuals only), indicating their potential antithrombotic effects in real life [60]. Another piece of evidence came from a study of carotid plaques after endarterectomy, showing that human atherosclerotic plaques readily incorporate EPA, which may render them less likely to trigger clinical events [61]. Based on this study, EPA and DHA could differ in their effects on membrane structure, rates of lipid oxidation, inflammatory biomarkers, endothelial function and tissue distributions. In summary, regarding the anti-inflammatory and antithrombotic properties of n-3 PUFAs, favorable effects were observed. Nevertheless, a huge spectrum of parameters studied in different experiments, including different species and different methodological approaches, precludes definite answers regarding exact biological mechanisms of n-3 PUFAs in this respect.

## 4. Cardiovascular Effects of n-3 PUFAs on (Sub)Cellular Level

As already indicated, DHA and EPA very probably differ in their clinical effects. Differences in their effect on clinically detectable factors such as plasma lipids and other markers of vascular damage are not striking and rather ambivalent. In this respect, interesting data regarding effects of DHA and EPA at the level of the cell membrane are at least partly explaining these differences. Sherratt et al. focused on models of the plasma membrane and observed strong relationships between hydrocarbon length and saturation with antioxidant activity and membrane cholesterol domain formation [62]. EPA had a favorable molecular structure contributing to membrane stability, improved lipoprotein clearance and reduced inflammation and reduced formation of radicals. In addition to ameliorating oxidative stress on the membrane, n-3 PUFAs could have direct effects on the structure of plasma membranes. A proposed mechanism by which n-3 PUFAs exert protective effects is their ability to significantly modulate cell membrane properties when incorporated into the phospholipid bilayer and to control membrane ion channels to prevent arrhythmias, including malignant ones (Figure 3).

The information regarding molecular targets of n-3 PUFAs is based mostly on experimental studies, including studies focused on Ca^2+^ and K^+^ ion channels. Interest was also raised in transient receptor potential vanilloid 4 (TRPV4) channels, which are Ca^2+^ channels mediating vasodilation. These channels play a role in decreasing systemic blood pressure (BP) by integrating hemodynamic forces and various metabolic inputs involved in endothelial and smooth muscle cells activation. TRPV4 promotes vasodilation through an increase of intracellular Ca^2+^, nitric oxide release and subsequent smooth muscle cell hyperpolarization. Interestingly, TRPV4 channels may be modulated by n-3 PUFA (but also by n-6 PUFAs) metabolites. Furthermore, n-3 PUFAs modulate TRPV4 function through plasma membrane remodeling. In vivo screening in a transgenic worm expressing rat TRPV4 demonstrated that EPA and its eicosanoid derivative, epoxyeicosatetraenoic acid (17,18-EEQ), are required for channel function. The potential clinical application of these findings is that n-3 PUFA-like molecules might also be developed as antihypertensive therapies targeting TRPV4 channels. This is further supported by the findings that in functional, lipid metabolome and biophysical analyses, n-3 PUFAs enhance TRPV4 function in human endothelial cells [63]. Another particular molecular target of n-3 PUFAs is potassium channels, which are important mainly in cerebrovascular responses, including BK_Ca_ channels and raising interest in the role of the potassium 2-pore channel, TREK-1. Cerebrovascular smooth muscle cells (CVSMCs) from TREK-1 knockout mice showed an enhanced response to 10^−4^ M ALA compared to wild-type mice with significant genotype-membrane potential interaction. It is interesting to note that ALA (non-specific activator of TREK-1) or arachidonic acid (activator of TREK-1) increased CVSMCs whole-cell currents, which were blocked by selective blockers of large-conductance Ca-activated K^+^ channels in both wild-type and TREK-1 (gene name KCNK2) knockout mice [64]. However, dilation of the cerebral artery induced by ALA was preserved in both strains, leading to conclusions that the TREK-1 current does not contribute to preserved vasodilator responses in the cerebral vasculature. Rather, PUFAs (both n-3 and n-6) exert their effects via BK_Ca_ channels [65] (Figure 4).

Regarding the potential effect of n-3 PUFAs on coronary arteries in experimentally studied endothelium-derived DHA products via lipoxygenase (17S-HDHA), activates BK_Ca_ channels in coronary arterial smooth muscle cells. In isolated perfused small bovine coronary arteries, 17S-HDHA (10^−9^ to 10^−5^ M) caused a concentration-dependent dilation with a maximum dilator response of 87.8 ± 2.5%, which is much more potent than the dilator response of its precursor, DHA. Moreover, vasodilatation induced by 17S-HDHA was significantly blocked by iberiotoxin, a large-conductance Ca^2+^-activated K^+^ (BK_Ca_) channel blocker, but not altered by an ATP-sensitive K^+^ channel blocker, glibenclamide [66]. Vasodilation mediated by endothelium-dependent hyperpolarization is reduced in hypertension by alterations of endothelial cell ion channels, but can be improved by antihypertensive therapy. These alterations include upregulation of endothelial Ca^2+^-activated chloride channels (CaClCs), functional loss of endothelial inward rectifier potassium (K_ir_) channels and downregulation of endothelial TRPV4 and SK_Ca_ channels [67]. Furthermore, EPA may directly or indirectly, through epoxygenase metabolites, modulate intracellular Ca^2+^ signaling in vascular smooth muscle cells by blocking Ca^2+^ current and inducing relaxation in the aorta of hypertensive rats [68]. It is interesting that DHA exhibits its relaxant action by affecting prostanoid receptor-mediated responses. The relaxant effects of DHA itself and its CYP epoxygenase metabolites on the rat aorta are mediated via K_ATP_ channels [44].

Regarding EPA, a very specific signaling pathway by which EPA exerts its effects on endothelial function in vivo was demonstrated in a study by Wu et al. [69]. The study was performed in mice deficient of either AMPKα1 or AMPKα2, in Apo-E/AMPKα1 dual KO mice, in models of bovine aortic endothelial cells and in eNOS knockout (KO) mice. EPA (0 to 100 µmol/L) induced endothelial nitric oxide synthase (eNOS) activation by activation of AMP-activated protein kinase via upregulation of mitochondrial uncoupling protein-2 (UCP-2). The final results were an increase in eNOS phosphorylation, an increase in NO release and consequent improvement of endothelial function in vivo. AMPKα1 or eNOS knockout mice fed a high-fat, high-cholesterol diet did not demonstrate favorable effects of EPA in their aortas, while genetic or pharmacologic inhibition of AMP-activated protein kinase abolished the beneficiary effects of EPA. Thus, it is evident that both AMPKα1 and NO are required for EPA-enhanced endothelium-dependent vasodilatation in ex vivo or in vivo.

In another experiment, spontaneously hypertensive rats were fed by fish oil diets containing 3% soybean oil and 4% tuna oil containing 27.8% DHA and 7% EPA [70]. The n-3:n-6 PUFA ratio was 1:9.5 for the control diet, whereas this ratio was reduced to approximately 1:1 for the fish oil diet. Rats were fed the diets for 12 weeks. The fish oil diet significantly reduced the mean arterial blood pressure, improved endothelial function and increased relaxation of carotid arteries in response to methacholine stimulation. An n-3 PUFA-rich diet reduced membrane arachidonic acid content, altered levels of specific (glucosyl)ceramide subspecies and decreased thromboxane concentrations in plasma. This was accompanied by a functional response, largely as reduced ceramide-induced contractions, mediated predominantly by thromboxane, suggesting that there was a suppression of sphingolipid-dependent vascular contractions due to n-3 PUFA intake.

Regarding pathophysiological processes in cardiac tissue, limited data are available that focus on the effects of n-3 PUFAs on energy metabolism. An interesting field is the phenomenon of ischemia-reperfusion injury [71]. Demaison et al. studied the effects of ischemia on heart pump function and mitochondrial energy metabolism conducted in male Wistar rats [72]. Rats were randomly assigned to diets containing a mixture of fish oil and sunflower seed or sunflower seed oil (both 100 g/kg). Results suggested that fish oil/n-3 PUFA-rich hearts recover their mitochondrial energy metabolism and myocardial pump functions better during reperfusion, possibly due to a higher resilience of n-3 PUFA-rich membranes to oxidative stress.

Regarding another area of intensive research in cardiology, namely heart failure with preserved ejection fraction (HFpEF), experimental studies suggest that EPA prevents interstitial fibrosis and diastolic dysfunction, potentially through EPA-free fatty acid receptor 4 (Ffar4) signaling. The findings in this study also indicated physico-chemical and biological diversity among n-3 PUFAs that influenced tissue distributions as well as disparate effects on membrane organization, rates of lipid oxidation, various receptor-mediated signal transduction pathways and gene expression. If confirmed in clinical settings, EPA could also be considered for the treatment of HFpEF [73] with a biological mechanism meant to be mediated through Ffar4, G protein-coupled receptor for n-3 PUFAs, which is able to block TGFβ1-fibrotic signaling in cultured cardiac fibroblasts, as previously reviewed [74]. Regarding human studies, interestingly and in certain contrast to the expected effects of EPA, in a retrospective single-center cohort study including 140 older patients with HFpEF, lower DHA and dihomo-gamma-linolenic acid levels, but not EPA or arachidonic acid levels, were significantly associated with an increase in all-cause mortality [75]. Moreover, in a multivariate regression analysis, low DHA levels were significantly associated with the incidence of all-cause mortality (HR: 0.16, 95% CI: 0.06–0.44), independently of the geriatric nutritional risk index. Therefore, low plasma levels of DHA may be a useful predictor of all-cause mortality in older patients with acute decompensated HFpEF. Nevertheless, it should be noted that the tissue concentration of particular n-3 PUFAs could be the major factor for given biological effects, and DHA needs not be a therapeutic target.

Because of the interplay between metabolic changes and cardiac function, including cardiac lipotoxicity, n-3 PUFAs through their direct (membrane) and indirect (lipid) effects could also be important players in the recently developing field of cardio-oncology in relation to lipid factors. In this respect, interesting data from experimental studies indicate the potential protective role of linoleic acid on cardiac structure in models of tumors [76,77].

## 5. Cardiovascular Effects of n-3 PUFAs in Studies of Subclinical Vascular Impairment

In addition to large clinical trials and experimental studies, studies focused on the combination of hemodynamic and metabolic parameters in individuals with particular cardiovascular characteristics could offer valuable insights into the multiple mechanisms of the effect of n-3 PUFAs, and generate interesting hypotheses before the completion of long-lasting, robust clinical studies. One of such studies is a randomized, controlled trial in young, healthy individuals supplemented with n-3 PUFAs (~1053 mg/per day for 3 weeks) through hen eggs, which demonstrated the effect of n-3 PUFAs on multiple levels, including a significant shift towards the production of anti-inflammatory prostanoids and increased levels of pro-resolving oxylipins [45,46,47,48]. Although both study and control groups showed reduced frequencies of peripheral nTreg lymphocytes and decreased rates of peripheral Th17 cells, their functional capacity for cytokine secretion was significantly altered only in the n-3 PUFAs group. The diet supplemented with n-3 PUFAs altered the immune response towards inflammation, thereby resolving conditions through effects on lipid mediators and cytokine secretion by T lymphocytes (reduced interleukin 6 secretions and increased transforming growth factor β-1).

Another trial, combining several investigative approaches was the OMEGA-PAD II trial, a double-blinded, randomized, placebo-controlled trial to assess the effects of 3 months of 4.4 g/day oral n-3 PUFA supplementation (in fish oil) on inflammation, endothelial function and walking ability in 24 patients with peripheral artery disease [78]. The absolute mean omega-3 index was significantly increased from baseline in patients who consumed fish oil (fish oil: 7.2 ± 1.2%, *p* < 0.001; placebo: −0.4 ± 0.9%, *p* = 0.31; between-group *p* < 0.001). The markers of specialized pre-resolvin mediators of biosynthesis, including several mono-hydroxyeicosapentaenoic acids and mono-hydroxydocosahexaenoic acids, were significantly increased. Namely, lipoxin A5 and resolving E3 were also increased. However, neither endothelium-dependent flow-mediated vasodilation of brachial artery nor a 6 min walk test were affected by “fish diet”. Moreover, C-reactive protein was measured by a high sensitivity method (hsCRP), and the results of the walking impairment questionnaire were not altered. Therefore, this study suggests more profound impacts of n-3 PUFAs on metabolic factors than on the vascular functional status in this group of high-risk patients with already-developed atherosclerotic disease. Different results were described in an open-label, single-blinded, randomized controlled study in 24 patients with coronary artery disease and impaired glucose metabolism evaluating consumption of 1.8 g EPA/day for 6 months [79]. Participants exhibited significant improvements in the EPA to arachidonic acid ratio, in fasting triglyceride levels and HDL-cholesterol concentration. The EPA group also exhibited a significant decrease in the incremental peak of the concentration of triglycerides, area under the curve (AUC) for postprandial triglycerides, incremental glucose peak, decrease of AUC for postprandial glucose and improvements in glycometabolism categorization. The EPA group also exhibited a significant increase in AUC-immune reactive insulin/AUC-plasma glucose ratio, which indicates postprandial insulin secretory ability. More importantly and in contrast to the previous study, in this study, flow-mediated dilation as a marker of endothelial function was significantly improved. Moreover, multiple regression analysis revealed that the flow-mediated dilation improvement was independently predicted by a decrease in the triglyceride to HDL-cholesterol ratio and incremental triglyceride peak in the EPA group. A similar beneficial effect of n-3 PUFAs on brachial artery flow-mediated dilation was also demonstrated in 45 patients with uncomplicated hypertension [80]. Additionally, an already mentioned, randomized-controlled trial [48] demonstrated that post-occlusive hyperemic response in microcirculation in young, healthy individuals was significantly enhanced, and triglycerides, hsCRP and BP were significantly decreased after consumption of functional food enriched with n-3 PUFAs; these effects were not observed in the control group, suggesting cardiovascular risk factors may be really attenuated by daily consumption of n-3 PUFAs enriched food. A recent cross-sectional study investigated the arterial BP-lowering effects of n-3 PUFAs in 2036 normotensive young and healthy individuals. Individuals in the highest quartile of the omega-3 index (EPA + DHA from red blood cells) had systolic BP and diastolic BP significantly lower than individuals in the lowest quartile (4 and 2 mmHg, respectively). In addition, a significant linear inverse relationship of the omega-3 index with 24-h and office BP was detected [81].

Finally, in the prospective cohort study of 2378 patients at very-high cardiovascular risk, with already established coronary artery disease (the Western Norway B-Vitamin Intervention Trial), available dietary data and available baseline glycosylated hemoglobin (HbA1c), reduced risk of acute myocardial infarction was associated with a high n-3 PUFA intake in diabetic patients (hazard ratio 0.38, 95% CI 0.18, 0.80) independently of HbA1c, while such association was not present in pre-diabetic patients [82]. However, an increased risk of fatal myocardial infarction was observed among patients with coronary heart disease without impaired glucose metabolism. Therefore, although these data were of an observational nature, some caution is to be considered in patients with coronary artery disease and normal glucose tolerance when they consume n-3 PUFAs at high doses. In summary, in most but not all studies of subclinical vascular impairment, positive effects of both EPA and DHA were demonstrated without EPA or DHA superiority.

## 6. Summary

The protective role of n-3 PUFAs on the vascular wall and myocardium stems from the wide spectrum of their indirect effects on metabolic, inflammatory and hemostatic factors, but also from their direct effects at the cellular level, particularly at the level of the cell membranes. The main and indirect metabolic effect of n-3 PUFAs on lipid metabolism is the decrease of plasma triglycerides mediated through modifying free fatty acid availability to hepatocytes, subsequent changes of VLDL metabolism and lowering atherogenic apoB-carrying lipoproteins and their protection from oxidation. In addition, the modulation of lipids includes various biologically active lipid mediators or modulation of systemic or vascular wall inflammatory conditions and oxidative stress, which could also indirectly improve vascular reactivity. Beyond the favorable effects on lipid metabolism, data regarding additional anti-inflammatory, antithrombotic and antiarrhythmic effects of n-3 PUFAs are promising (Table 2).

Regarding vascular and cardiac effects, differences exist between two main representatives of n-3 PUFAs, DHA and EPA. EPA seems to be more efficient in the favorable mitigation of the atherosclerotic process due to its molecular properties improving oxidative processes at the level of the cell membrane. Regarding the effects of n-3 PUFAs on ion channels and progression of HFpEF, favorable effects were also detected, but in this case, no clear advantage of DHA or EPA was observed; different pathophysiological mechanisms could be responsible (Figure 5).

In addition, (epoxy)metabolites of n-3 PUFAs can significantly modify vascular/cellular function, and attention should be focused on this field. However, it should be kept in mind that detailed descriptions of specific pathophysiological properties of n-3 PUFAs are based mostly on in vitro dose-dependent acute effects, often using genetically altered animal models. Moreover, direct n-3 PUFA effects on vascular function in experimental studies are investigated mostly in pulmonary circulation (e.g., in pulmonary hypertension models), in coronary microcirculation and in hypertensive animal models (e.g., spontaneously hypertensive rats, knockout mice, etc.).

Despite evidence of the successful prevention of cardiovascular disease by relatively high doses of icosapent ethyl (derivative of EPA) in one large randomized trial, but negative results in other trials focused on this treatment, there is still a need for confirmatory studies. Treatment duration and/or timing of treatment might be of primary importance in the effective prevention of pathological vascular changes. Further investigation of the exact mechanisms of n-3 PUFA effects on a molecular level is definitely needed.

## Figures and Tables

**Figure 1 ijms-23-02104-f001:**
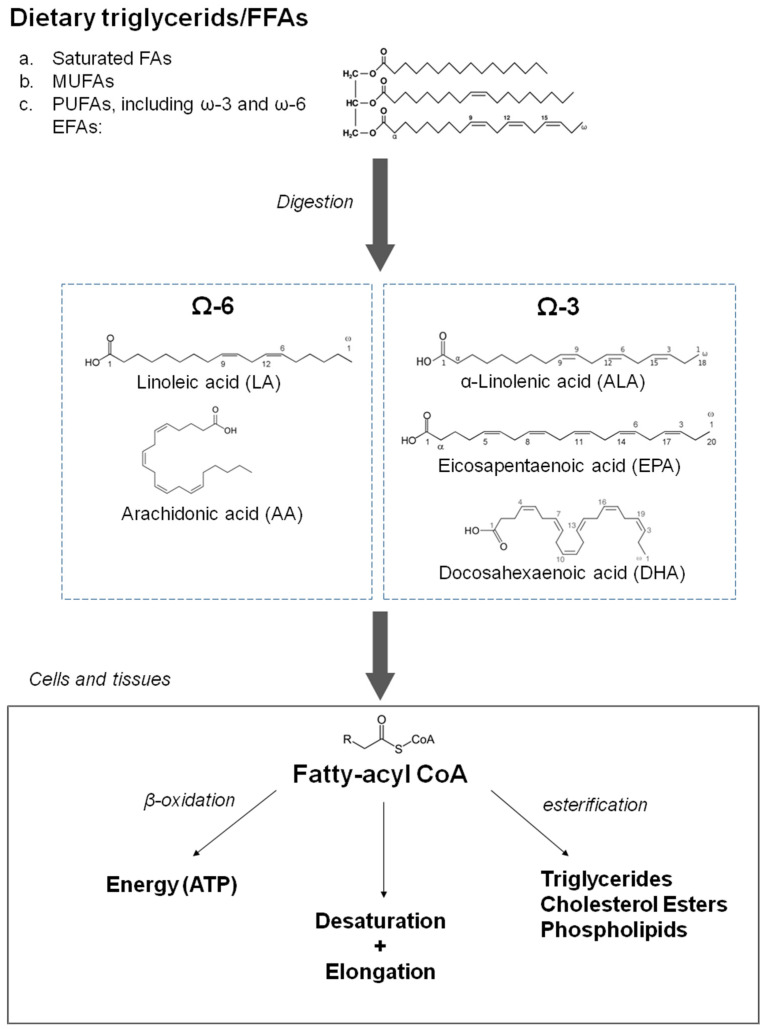
Characteristics of dietary fatty acids with a focus on omega-3 fatty acids and related compounds (source: Drenjančević, Ines et al. “The Effect of Dietary Intake of Omega-3 Polyunsaturated Fatty Acids on Cardiovascular Health: Revealing Potentials of Functional Food”. Superfood and Functional Food—The Development of Superfoods and Their Roles as Medicine, edited by Naofumi Shiomi, Viduranga Waisundara, IntechOpen, 2017. 10.5772/67033).

**Figure 2 ijms-23-02104-f002:**
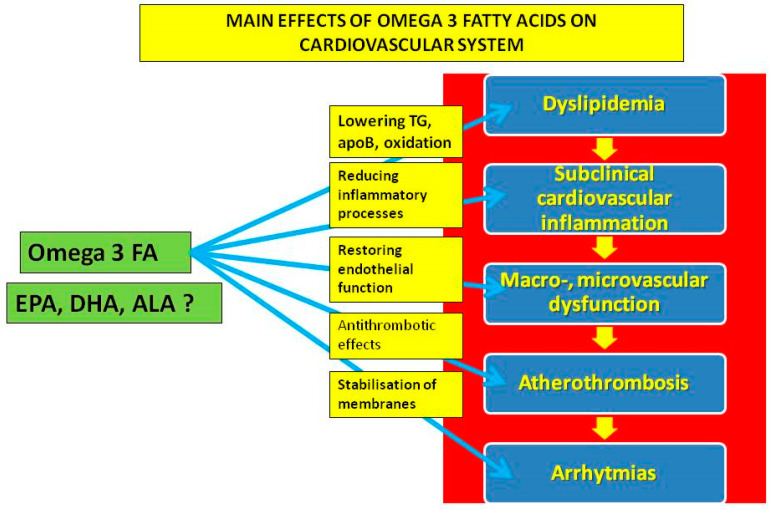
The main potential effects of omega-3 fatty acids on the cardiovascular system.

**Figure 3 ijms-23-02104-f003:**
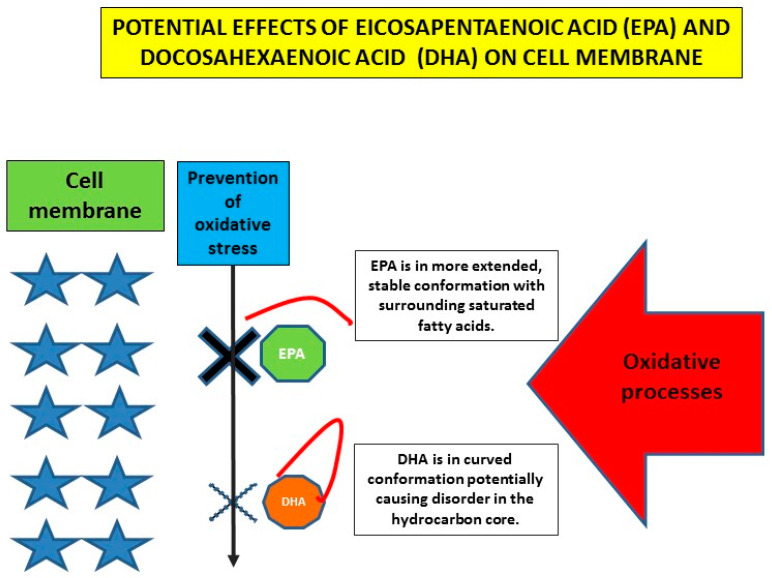
Schematic representation of potential difference between EPA and DHA at the level of the plasma membrane (created partly on information and data from [39,62]).

**Figure 4 ijms-23-02104-f004:**
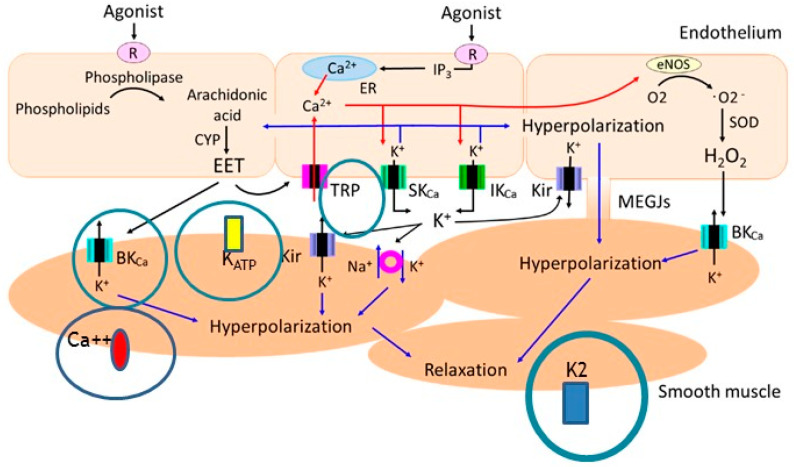
The ion channels affected by omega-3 fatty acids + metabolites important in the mechanisms of vascular relaxation with a focus on K2 channels (figure adopted from Kenichi Goto, Toshio Ohtsubo and Takanari Kitazono. Endothelium-Dependent Hyperpolarization (EDH) in Hypertension: The Role of Endothelial Ion Channels. Int. J. Mol. Sci. 2018, 19(1), 315; https://doi.org/10.3390/ijms19010315, accessed on 10 February 2022 and modified by I.D.) [65].

**Figure 5 ijms-23-02104-f005:**
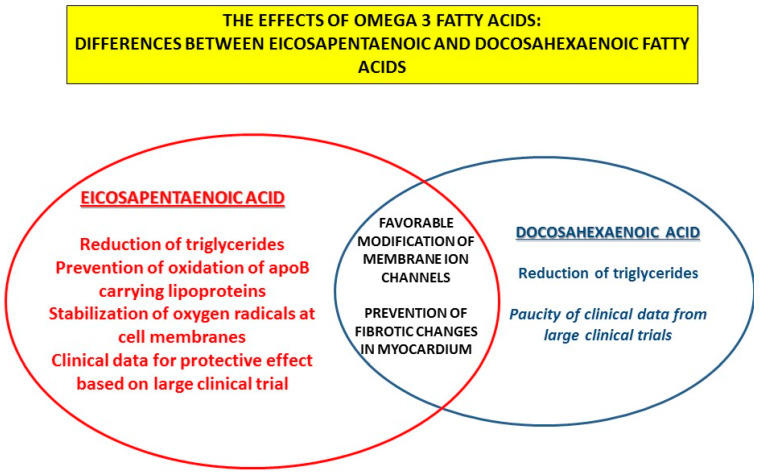
The effects of omega-3 fatty acids: differences and similarities between eicosapentaenoic and docosahexaenoic fatty acids.

**Table 1 ijms-23-02104-t001:** Main clinical trials focused on the effect of n-3 PUFAs on cardiovascular events (created partly on information and data from [25]).

	Type of Intervention	Main Characteristics	Clinical Effect Including Adverse Events
Successful—primary endpoint reached			
REDUCE-IT [8]	Icosapent ethyl 4 g/day, mineral oil as control	Patients with established CVD or DM on statin therapy with increased TG (*n* = 8179; 29% women, FU 4.9 years), median TG 216 mg/dL (2.44 mmol/L)	The risk of CV death, MI, revascularization, unstable angina was significantly lower in treated patients (26% reduction). The incidence of atrial fibrillation and peripheral edema were significantly higher in the treatment group. The overall rates of serious adverse bleeding were similar. No fatal bleedings and no between-group differences in the rates of hemorrhagic stroke, serious central nervous system bleeding, or gastrointestinal bleeding.
EVAPORATE [9]	Icosapent ethyl 4 g/day, mineral oil as control	Patients with confirmed coronary artery stenosis, on statin therapy with increased TG (*n* = 80; 44 women, FU 1.5 year), median TG 191–200 mg/dL (2.48–2.60 mmol/L)	Low-attenuation plaque volume and thickening of fibrotic cap were significantly reduced in the treatment group. No serious adverse effects were described.
JELIS [10]	EPA 1.8 g/day (+pravastatin or simvastatin), no placebo	Patients with previous MI or PCI or with confirmed angina pectoris or without CVD (*n* = 18,645; 69% women, FU 4.6 years), TG 150.4 mg/dL (1.7 mmol/L)	All-cause mortality was reduced in secondary, but not in primary prevention. Adverse experiences and discontinuation rate due to treatment was more common in the treatment group, including laboratory data, gastrointestinal disturbances, skin abnormalities, and cerebral and fundal bleedings, epistaxis and subcutaneous bleeding (all mild). Incidence of new cancers was not different between groups.
CHERRY [26]	Pitavastatin and EPA therapy (4 mg/day and 1800 mg/day) vs. pitavastatin (4 mg/day),	Patients after PCI, pitavastatin/EPA group (*n* = 97) vs. pitavastatin group (*n* = 96), 16–20% women, FU 6–8 months, median TG 105–111 mg/dL (1.2–1.25 mmol/L)	Statin + EPA therapy significantly reduced coronary plaque volume vs. statin therapy alone. Plaque stabilization was reinforced by statin/EPA in patients with stable angina pectoris. No significant difference in adverse events, including atrial fibrillation.
Not successful			
ORIGIN [27]	EPA (465 mg) + DHA (375 mg) vs. placebo (approx. 1 g of olive oil).	High risk of CVD + impaired Fasting glucose/glucose intolerance/DM (*n* = 12,536; 35% women; FU 6.2 years). Median TG 140–142 mg/dL (1.58–1.60 mmol/L)	No reduction in non-fatal MI or stroke, death from CV cause or arrhythmia was observed. No differences in major bleeding.
ASCEND [11]	EPA + DHA combined (1 g/day) vs. olive oil capsule	Persons older than 40 years + DM, without CVD (*n* = 15,480, FU 7.4 years, 37% female), TG not measured	No effect on non-fatal MI or stroke, vascular death in treated patients. No significant between-group differences in the rates of non-fatal serious adverse events, including new cancers.
STRENGTH [12]	EPA + DHA combined (4 g/day) vs. corn oil.	Patients with CVD or at high risk for CVD (*n* = 13,078; FU 3.5 years, 35% women), median TG 240 mg (2.72 mmol/L)	No significant effect of treatment on CV death, non-fatal MI, stroke, coronary revascularization or unstable angina observed. The incidence of gastrointestinal adverse events was higher in the treatment group. New-onset atrial fibrillation was more common in the treatment group. Major and minor bleeding events were not different between groups.
OMEMI [14]	EPA 930 mg + DHA 660 mg vs. corn oil	Age 70 to 82 year + recent (2–8 weeks) acute MI (*n* = 1014, FU 2 years; 29% females), mean TG 111.4 mg/dL (1.26 mmol/L)	No reduction in clinical events. No differences in major bleeding (10.7% vs. 11.0% in controls). No patients withdrew because of bleeding problems. Reasons for discontinuing treatment not different between the groups (gastrointestinal symptoms, difficulty swallowing capsules, other disease burdens not related to the study intervention).
VITAL [28]	Vitamin D3 (2000 IU per day) and 1 g fish-oil capsule (EPA-460 mg, DHA-380 mg) vs. placebo (not specified)	Healthy men > 50 and women > 55 years of age (*n* = 25,871; FU 5.3 years, 51% women), TG not specified	No effect on MI, stroke, CV death in treated patients was detected; benefit was observed in African Americans (HR: 0.23); PCI (HR: 0.78); fatal MI (HR: 0.50). No excess risks of bleeding or other serious adverse events, including gastrointestinal symptoms, major bleeding episodes observed

Legend: AF: atrial fibrillation, CV: cardiovascular, CVD: cardiovascular disease, DHA: docosahexaenoic acid, DM: diabetes mellitus, EPA: eicosapentaenoic acid, FU: follow up, HR: hazard ration/CI: confidence intervals, MI: myocardial infarction, PCI: percutaneous coronary intervention.

**Table 2 ijms-23-02104-t002:** Potential effects of omega-3 fatty acids on cardiovascular system.

Main Targets	Mechanisms	Evidence	Clinical Effect
Indirect effects: Circulating free fatty acids Circulating triglycerides Thrombotic factors Inflammatory factors	Free fatty acid availability Reduction of subclinical inflammation, detectable by measurement of the wide spectrum of inflammatory markers, anti-fibrotic effects	One successful randomized trial with icosapent ethyl (EPA derivative) in higher dose, parallel trial with reduction of coronary atherosclerotic plaques in DHA data less convincing, potential role in HFpEF (evidence from observational study based on DHA plasma levels).	Lowering plasma triglycerides Lowering blood pressure Improving endothelial function measured in peripheral vessels (mostly experimental settings in humans)
Direct effects: Cell membranes Ion channels	Membrane stabilization, including their protection against free radicals Modification of different types of ion channels	In human, mostly indirect evidence from experiments, in vitro studies. Observational studies indicating decrease of sudden death with intake of fish oil.	Experimental data In vitro studies Observational/epidemiological studies in humans

Legend: DHA: docosahexaenoic acid; EPA: eicosapentaenoic acid HFpEF: heart failure with preserved ejection fraction.

## Data Availability

Not applicable.

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
