# Peer review of "Omega-3 Polyunsaturated Fatty Acids—Vascular and Cardiac Effects on the Cellular and Molecular Level (Narrative Review)"

_ijms, 2022, doi:10.3390/ijms23042104_

Round 1
Reviewer 1 Report
Manuscript was significantly improved according to the Reviewers' suggestion.
Author Response
Thank you, we did our best to further improve language, fluency and clarity to the manuscript.
Reviewer 2 Report
ijms-1601259, Omega -3 Polyunsaturated Fatty Acids - Vascular and Cardiac Effects on the Cellular and Molecular Level (Narrative Review)
The manuscript is a resubmission of the ijms-1502746. The authors improved their paper based on the previous comments. I advise the authors to add a scheme with the chemical structures of α-linolenic acid (ALA), eicosapentaenoic acid, (EPA), and docosahexaenoic acid (DHA). The manuscript has a good scientific value and could be interesting for the journal’s readers.
Author Response
Thank you,
we added the scheme (Figure 1) and did necessary changes plus we improved the language, fluency and clarity of our paper.
This manuscript is a resubmission of an earlier submission. The following is a list of the peer review reports and author responses from that submission.
Round 1
Reviewer 1 Report
ijms-1502746, Omega -3 Polyunsaturated Fatty Acids - Vascular and Cardiac Effects on the Cellular and Molecular Level (Narrative Review),
The manuscript is a resubmission of the ijms-1338985. The authors improved their paper based on the previous comments of the review.
The authors detailed on the types of fatty acid and the meaning of n-3, PUFA, or omega-3, but I think that a scheme with the structures of α-linolenic acid (ALA), eicosapentaenoic acid, (EPA), and docosahexaenoic acid (DHA).
The authors added the doses that produce the effects described, but there are still missing data that should be added. For example, on row 218 and row 273.
In the Conflicts of Interest section, please mention if the lecture honoraria from pharmaceutical companies involved Omega -3 Polyunsaturated Fatty Acids.
Reviewer 2 Report
Drenjancevic and Pitha provide a narrative review on vascular and cardiac effects of omega-3 fatty acids at the cellular and molecular level.
Major Points
On December 8, a medline search with “omega-3” resulted in 34 282 publications, with most of them focusing on mechanisms. In the abstract, the authors deplore “a paucity of data describing exact mechanisms of n-3 PUFA…”. This way, the authors rather demonstrate that they are not conversant with the field of omega-3’s. Unfortunately, the same is true for cardiovascular disease, since the authors state in Introduction – without reference – that depression is an “established cardiovascular risk factor”, which it is not. Moreover, it is unclear, what “depression” the authors are talking about, illustrating their being unfamiliar with psychiatric nomenclature. Unwittingly, the authors go on to prove my point by stating that humans “can use ALA, when available, to form EPA and DHA.” Humans cannot form DHA from ALA in sufficient amounts, as demonstrated in JELIS, among other trials (e.g. PMID’s 21099130, 29459911, but many others). The statements to follow are further evidence to prove my point.
Minor Points
Language needs improvement. Many sentences are too long, with the first two sentences in the abstract being examples. Many expressions are unusual. I suggest Editing.
Reviewer 3 Report
The manuscript numbered ijms-1502746 deals with the vascular and cardiac effect of n-3 PUFA on the cellular and molecular level.
My greatest concern is the lack of novelty – how this research expand existing knowledge or bring any new knowledge to the medicine, food science, etc.? Better justification of this review is required. Although the outline of the general need is given, more logical explanation should be presented. This review should be better structured. Authors definitely should divide described evidences and trials on animal and human studies. They may focus on n-3 PUFA from food sources (including dietary supplements) and describe how these FA influence selected factors/parameters when used by healthy people or by patients with CVD, when used singly or jointly with pharmacotherapy.
Manuscript should be subjected to major revision and then resubmitted to International Journal of Molecular Sciences. Detailed remarks are presented below.
- Lines 50 - 56: Please use the proper notation for fatty acid names, e.g. for ALA it should be: cis9cis12cis15 C18:3. You may also use only ‘c’ as abbreviation for ‘cis’. Remember to use italics for both ‘cis’ and ‘c’ as it came from Latin.
- Lines 57 – 58: Please give reference to the information that n-3 PUFA are resistant for oxidation.
- Lines 60 – 61: You may use only EPA and DHA as their whole names was given earlier.
- Line 96: What type of plant products are considered? Are they dietary sources of n-3 PUFA? What type of n-3 PUFA are taken into account? All of these FA or especially long-chain? If so, plant materials are not a good sources of EPA and DHA, as these FA are present mainly in fish, fish products, e.g. fish oil. Such important issues should be clarified and clearly stated here.
- Line 102 – 105: This paragraph should be given earlier. Here it is not suitable.
- Better justification of this review is required. Although the outline of the general need is given, more logical and ‘cause and effect’ explanation should be presented. Why this review is so important? How it expand existing knowledge? Aspects of novelty should also be emphasized.
- The information how lipids, especially FA, are important for heart are lacking in this section. Please provide such information together with the statement that abnormalities in lipid metabolism may be responsible for cardiac and cardiovascular health deterioration. Recently some papers focused on this topic were published: Nutrients, 2019, 11, 9, 2032; Animals, 2020, 10 (3), 464; Chemistry and Physics of Lipids, 2021, 235, 105057; Journal of Trace Elements in Medicine and Biology, 2021, 68, 126816; Molecules, 2021, 26 (23), 7127 – You may use them as a reference.
- Line 140: In what material: whole blood, plasma, serum or other? Please specify.
- Whole manuscript should be carefully checked by the native speaker as some language mistakes occurs.